# Biofilm Formation among *Stenotrophomonas maltophilia* Isolates Has Clinical Relevance: The ANSELM Prospective Multicenter Study

**DOI:** 10.3390/microorganisms9010049

**Published:** 2020-12-27

**Authors:** Arianna Pompilio, Marco Ranalli, Alessandra Piccirilli, Mariagrazia Perilli, Dragana Vukovic, Branislava Savic, Marcela Krutova, Pavel Drevinek, Daniel Jonas, Ersilia V. Fiscarelli, Vanessa Tuccio Guarna Assanti, María M. Tavío, Fernando Artiles, Giovanni Di Bonaventura

**Affiliations:** 1Laboratory of Clinical Microbiology, Department of Medical, Oral and Biotechnological Sciences, “G. d’Annunzio” University of Chieti-Pescara, 66100 Chieti, Italy; marco.ranalli91@gmail.com (M.R.); gdibonaventura@unich.it (G.D.B.); 2Operative Unit of Clinical Microbiology, Center for Advanced Studies and Technology (CAST), “G. d’Annunzio” University of Chieti-Pescara, 66100 Chieti, Italy; 3Department of Biotechnological and Applied Clinical Sciences, Health and Environmental Sciences, University of L’Aquila, 67100 L’Aquila, Italy; alessandra.piccirilli@univaq.it (A.P.); mariagrazia.perilli@univaq.it (M.P.); 4Institute of Microbiology and Immunology, Faculty of Medicine, University of Belgrade, 11000 Belgrade, Serbia; dragana.vukovic@med.bg.ac.rs (D.V.); branislava.savic@med.bg.ac.rs (B.S.); 5Department of Medical Microbiology, 2nd Faculty of Medicine and Motol University Hospital, Charles University, 15006 Prague, Czech Republic; marcela.krutova@lfmotol.cuni.cz (M.K.); pavel.drevinek@lfmotol.cuni.cz (P.D.); 6University Freiburg—Medical Center, Institute for Infection Prevention and Hospital Epidemiology, 79098 Freiburg, Germany; daniel.jonas@uniklinik-freiburg.de; 7Laboratory of Cystic Fibrosis Microbiology, “Bambino Gesù” Children’s Hospital IRCCS, 00165 Rome, Italy; evita.fiscarelli@opbg.net (E.V.F.); vanessa.tuccio@opbg.net (V.T.G.A.); 8Microbiology, Clinical Science Department, Faculty of Health Sciences, University of Las Palmas de Gran Canaria, 35001 Las Palmas de Gran Canaria, Spain; mariadelmar.tavio@ulpgc.es; 9Microbiology & Parasitology Service, University Hospital of Gran Canaria Dr. Negrín, 35001 Las Palmas de Gran Canaria, Spain; fartcam@gobiernodecanarias.org

**Keywords:** *Stenotrophomonas maltophilia*, biofilm formation, clinical relevance, antibiotic resistance, multicenter study

## Abstract

The ability to form biofilms is a recognized trait of *Stenotrophomonas maltophilia*, but the extent of its clinical relevance is still unclear. The present multicenter prospective study (ANSELM) aims at investigating the association between biofilm formation and clinical outcomes of *S. maltophilia* infections. One hundred and nine isolates were collected from various geographical origins and stratified according to their clinical relevance. Biofilm formation was evaluated by the microtiter plate assay and correlated with microbiological and clinical data from the associated strains. Antibiotic susceptibility of the planktonic cells was tested by the disk diffusion technique, while antibiotic activity against mature biofilms was spectrophotometrically assessed. Most strains (91.7%) were able to form biofilm, although bloodborne strains produced biofilm amounts significantly higher than strains causing hospital- rather than community-acquired infections, and those recognized as “definite” pathogens. Biofilm formation efficiency was positively correlated with mechanical ventilation (*p* = 0.032), whereas a negative relationship was found with antibiotic resistance (*r*^2^ = 0.107; *p* < 0.001), specifically in the case of the pathogenic strains. Mature *S. maltophilia* biofilms were markedly more resistant (up to 128 times) to cotrimoxazole and levofloxacin compared with their planktonic counterparts, especially in the case of bloodborne strains. Our findings indicate that biofilm formation by *S. maltophilia* is obviously a contributing factor in the pathogenesis of infections, especially in deep ones, thus warranting additional studies with larger cohort of patients and isolates.

## 1. Introduction

*Stenotrophomonas maltophilia* is a globally emerging multidrug-resistant Gram-negative pathogen. It has a propensity to cause a plethora of opportunistic infections in humans, mainly associated with the respiratory tract [1].

In addition to the antibiotic resistance, another significant trait of *S. maltophilia* virulence is the ability to adhere to abiotic surfaces (e.g., respiratory tubes, intravenous cannulae, prosthetic devices, dental unit waterlines, and nebulizers) and host tissues (e.g., HEp-2 monolayers, lung cells, tracheal cells) where it grows as a biofilm forming sessile communities that are inherently resistant both to antibiotic therapy and the host immune response [1,2,3,4,5,6,7,8,9,10,11,12].

Recently, a handful of studies have reported the frequency and characterization of biofilm-producing *S. maltophilia* strains prospectively isolated from several hospitals worldwide. Overall, the findings revealed that biofilm formation is highly conserved in *S. maltophilia* and occurs with relevant efficacy leading to high biomass amount [2,5,13,14,15,16,17,18,19,20,21]. However, these studies were aimed at investigating the phenotypic and genotypic characteristics or virulence traits, without providing evidence for the relationship between biofilm formation and the clinical course of diseases.

In the present study, 109 *S. maltophilia* clinical isolates were collected during a multicenter prospective cohort study involving five European hospitals and were evaluated for biofilm formation, antibiotic resistance, and genetic heterogeneity. In addition, the efficiency of biofilm formation and the biofilm resistance to commonly used antibiotics were cross-referenced both with microbiological and clinical data aimed at determining possible relationships. The knowledge gained from these results may contribute to the design of novel diagnostic and therapeutic interventions to prevent and/or cure biofilm-associated infections caused by *S. maltophilia*.

## 2. Materials and Methods

This study (ANSELM, clinicAl sigNificance of *StEnotrophomonas maLtophilia* biofilM) was approved by the Ethics Committee of “G. d’Annunzio” University of Chieti-Pescara, Italy (permission number 1864, 11 December 2018).

### 2.1. Bacterial Strains

A total of 109 *S. maltophilia* isolates were collected from different clinical specimens at selected institutions in five European countries between February–July 2019.

One isolate per patient was included in the study. Species identification was performed using a miniaturized BD BBL Crystal system (Becton, Dickinson and Company, Franklin Lakes, NJ, USA) or an automated system (Vitek 2 or Vitek MS system, bioMérieux SA, F-69280 Marcy l’Etoile, France; BD Phoenix 100, Becton, Dickinson and Company). Each isolate was stored at −80 °C in a Microbank™ cryogenic system (Biolife Italiana, Milan, Italy) until use when it was plated three times onto Müller-Hinton Agar (MHA; Oxoid SpA, Garbagnate M.se, Milan, Italy) to recover the original phenotypic traits.

### 2.2. Multilocus Sequence Typing (MLST)

MLST was based on the sequence data of seven housekeeping genes: *tpD* (H(+)-transporting two-sector ATPase); *gapA* (NAD-dependent glyceraldehyde-3-phosphate dehydrogenase); *guaA* (GMP synthase [glutamine-hydrolyzing]); *mutM* (DNA-formamidopyrimidine glycosylase); *nuoD* (NADH dehydrogenase (ubiquinone)); *ppsA* (pyruvate, water dikinase); and *recA* (RecA protein). Gene amplification was performed by PCR according to a standardized protocol (https://pubmlst.org/smaltophilia/info/primers.shtml). Genomic bacterial DNA was extracted using the Euroclone spinNAker Universal Genomic DNA mini kit (Euroclone, Milan, Italy). The positive amplicons derived from three different PCR reactions were sequenced on both strands by BigDye Sequencing Reaction Kit using an ABI PRISM 310 capillary automated sequencer (Applied Biosystem, Monza, Italy). Alleles and sequence types (STs) are accessible online (https://pubmlst.org/bigsdb?db=pubmlst_smaltophilia_seqdef).

### 2.3. Microbiological and Clinical Data

For each strain, the following microbiological and clinical data were collected by each Center participating in the study: (i) age and gender of the patient; (ii) isolation site; (iii) community-acquired infection (if it occurred at least 48 h prior to hospitalization) or hospital-acquired infection (if it occurred at least 48 h after admission); (iv) ward, if hospital-acquired infection; (v) clinical presentation; (vi) clinical diagnosis; vii) underlying comorbidities; (viii) risk factors; (ix) clinical outcome; and (x) antibiotic therapy, indicating drugs, dose, administration route, and duration of both “empiric” therapy (started before *S. maltophilia* isolation) and “targeted” therapy (that is a drug administrated based on the results of antimicrobial susceptibility testing).

The etiological role of each strain was defined according to the CDC guidelines [22] and were as follows: (i) “definite pathogen”, if the patient had symptoms and signs of infection at the site of isolation and no other pathogen was isolated from that site; (ii) “probable pathogen”, if the patient had symptoms and signs of infection at the site of isolation but the culture yielded polymicrobial growth; (iii) “possible pathogen”, if the signs and symptoms of infection were evident but not clearly related to the site of isolation; (iv) “nonpathogen”, if there was no evidence of infection at the time of isolation.

### 2.4. Standardized Inoculum Preparation

A standardized inoculum was prepared according to the intended use.

For biofilm formation assay and microscopic observation, several colonies were grown overnight at 37 °C onto MHA, resuspended in 5 mL Trypticase Soy Broth (TSB; Oxoid SpA), and then incubated at 37 °C under agitation (130 rpm). After overnight incubation, the broth culture was corrected with sterile TSB to an optical density measured at 550 nm (OD_550_) of 1.0 (corresponding to 1–2 × 10^9^ CFU/mL), and then diluted 1:100 (*v*/*v*) in TSB to achieve a final inoculum concentration of 1–2 × 10^7^ CFU/mL.

For antibiotic susceptibility tests (both of planktonic and biofilm cells), a suspension of several overnight MHA growth colonies prepared in 5 mL of sterile saline 0.9% (Fresenius Kabi, Verona, Italy) was corrected at an OD_550_ of 0.1 (corresponding to 1–2 × 10^8^ CFU/mL), and then diluted 1:10 (*v*/*v*) in Cation-Adjusted Müller-Hinton Broth (CAMHB; Becton, Dickinson and Company).

Inoculum size and purity were checked by a viable cell count on MHA.

### 2.5. Microtiter Plate (MTP) Assay for Biofilm Quantification

Biofilm biomass—including both adherent bacteria and extracellular polymeric substance (EPS)—was measured by crystal violet MTP assay. In brief, standardized inoculum (200 μL) was aseptically added to each well of a 96-well polystyrene tissue culture plate (Becton, Dickinson and Company) and incubated aerobically at 37 °C under static conditions. Wells containing TSB only were considered as controls. At the end of the incubation, spent medium was discarded and each well was washed twice with PBS (pH 7.2) (Sigma-Aldrich, Milan, Italy) to remove non-adherent cells. Biofilm samples were fixed by incubating plates at 60 °C for 1 h, and then stained for 5 min with Hucker-modified crystal violet (200 μL) [23]. After the plates were air-dried, crystal violet was extracted by exposure for 15 min to 200 μL of 33% glacial acetic acid (Sigma-Aldrich), and the biofilm biomass was then assessed by measuring the optical density at 492 nm (OD_492_) (Sunrise, Tecan, Milan, Italy). According to the criteria proposed by Stepanović et al. [24], each strain was classified for biofilm formation efficiency as follows: (i) non-producer (OD ≤ ODc); weak-producer [ODc < OD ≤ (2 × ODc)]; moderate-producer [(2 × ODc) < OD ≤ (4 × ODc)]; and strong-producer (OD > 4 × ODc). Cut-off value (ODc) was defined as OD (mean negative control) + 3 × standard deviations.

### 2.6. Antibiotic Susceptibility of Planktonic Cells

The susceptibility of *S. maltophilia* strains to ceftazidime (CAZ), chloramphenicol (CHL), colistin (CST), trimethoprim-sulfamethoxazole (SXT), levofloxacin (LVX), minocycline (MIN), and ticarcillin-clavulanic acid (TIM) was evaluated by the disk diffusion technique, according to the CLSI guidelines [25].

MIC values for SXT and LVX were obtained by the broth microdilution technique, according to the CLSI guidelines [25]. MBC was evaluated in duplicate by culture; a 10 μL media from wells showing no visible growth at MIC determination were inoculated onto MHA. Following incubation at 37 °C for 24 h, an MBC value was defined as the minimum antibiotic concentration able to eradicate 99.9% of the starting inoculum.

*Escherichia coli* ATCC25922 and *Pseudomonas aeruginosa* ATCC27853 reference control strains were assessed in parallel for quality control.

### 2.7. Antibiotic Activity Against Biofilm Formation and Mature Biofilm

The Minimum Biofilm Inhibitory Concentration (MBIC) and the Minimum Biofilm Eradication Concentration (MBEC) values of LVX and SXT were assessed against a *S. maltophilia* biofilm. An aliquot (200 μL) of the standardized inoculum was added into each well of a 96-well polystyrene, flat bottom, tissue culture-treated microtiter (Iwaki, Bibby srl; Milan, Italy) and aerobically incubated at 37 °C for 24 h under static conditions. Biofilm samples were washed once with PBS (200 µL) and then exposed to each tested antibiotic at several concentrations previously prepared in CAMHB.

Control samples were also prepared as follows: (i) biofilm exposed to broth only (killing activity: 0%); and biofilm exposed to 100% DMSO (killing activity: 100%).

After a 20 h-exposure to antibiotics, biofilm samples were washed twice with sterile PBS (200 µL) and then TSB (200 µL) was added to each well to verify their effect on the biofilm. After 6 h- and 24 h-incubation at 37 °C, OD_620_ of broth supernatant was spectrophotometrically assessed. MBIC was defined as the lowest antibiotic concentration allowing a regrowth of ≤10% compared to the positive (unexposed) control well readings (representing at least a 1-Log growth difference), whereas MBEC was defined as the lowest antibiotic concentration causing biofilm eradication (i.e., no growth, like negative controls exposed to DMSO).

### 2.8. Evaluation of Biofilms by Microscopic Analysis

The ultrastructure of the biofilm produced by selectedrepresentative *S. maltophilia* strains was evaluated by both Scanning Electron Microscopy (SEM) and Confocal Laser Scanning Microscopy (CLSM).

SEM analysis: biofilm sample grown (37 °C, 24 h, static) on a polystyrene coupon accommodated in a 6-well microtiter (Becton, Dickinson and Company) was washed twice in PBS, then fixed overnight in a mixture of 2% paraformaldehyde (Electron Microscopy Sciences, Rome, Italy) + 2% glutaraldehyde (Sigma-Aldrich) in 0.15 M sodium cacodylate buffer (pH 7.4; Honeywell Fluka, Milan, Italy), with the addition of alcian blue 0.1% (8GX; Sigma-Aldrich) aimed at detecting EPS. Samples were post-fixed for 90 min in 1% OsO_4_ (Electron Microscopy Sciences) in 0.15 M cacodylate buffer, dehydrated in ascending ethanol series, stained en-bloc with 2% alcoholic uranyl acetate for 60 min and rinsed in 100% ethanol. Samples were then gold coated in a Desk Sputter Coater 1 (PVD, Teheran, Iran), and finally analysed with the Phenom Pro (Thermo Fisher Scientific, Monza, Italy) under “high-vacuum” modality and with an operating voltage of 15 kV.

CLSM analysis: biofilm samples grew (37 °C, 24 h, static) onto TC-treated, 35 mm diameter, μ-Dish (Ibidi, Milan, Italy) were washed once with PBS, and then were stained (15 min, room temperature) with Live/Dead BacLight kit (Molecular Probes, Milan, Italy) to assess the viability, along with Concanavalin A (ConA, Alexa Fluor 647 Conjugate; Molecular Probes) to detect EPS (both from Thermo Fisher Scientific). CLSM analysis was performed with an LSM 510 META laser scanning microscope attached to an Axioplan II microscope (Zeiss Italia, Arese, Milan, Italy).

Representative images were acquired during both SEM (ProSuite software Thermo Fisher Scientific) and CLSM (ZEN 2.3 SP1 software, Carl Zeiss, ver. 14.0) observations and processed for display using Photoshop (Adobe Systems Inc., San Jose, CA, USA) software.

### 2.9. Statistical Analysis

Each experiment was carried out at least in triplicate and repeated on two different occasions (*n* ≥ 6). The distribution of results was assessed using a D’Agostino-Pearson normality test, and then the differences in the biofilm biomass (OD_492_) were evaluated accordingly: (i) using a Kruskal-Wallis test followed by a Dunn’s multiple comparisons post-test (among more than 2 groups) or a Mann-Whitney test (between 2 groups), in case datasets did not pass the normality test; (ii) by an unpaired-t test, in cases of normally distributed datasets. The statistical significance of differences between variables was calculated by a Chi-square test. The relationship between biofilm biomass and antibiotic resistance level was assessed by linear regression analysis. MBIC/MIC and MBEC/MBC values were considered statistically significant if >2. The significance level was set at *p* < 0.05.

## 3. Results

### 3.1. Microbiological and Clinical Features

The microbiological and clinical characteristics of the 109 strains and patients enrolled in the present study were stratified by the *S. maltophilia* etiological role, according to the CDC guidelines (Table 1). A total of 76 strains (69.7%) was found to be etiologically relevant having been classified as a “definite” (*n* = 23; 21.1%), “probable” (*n* = 45; 41.3%) or “possible” (*n* = 8; 7.3%) pathogen. The remaining 33 strains (30.3%) were classified as “non-pathogen”.

Considering the strains as a whole, the airways were the most prevalent isolation site (*n* = 64; 58.7%, *p* < 0.0001 vs. other groups), followed by blood (*n* = 23; 21.1%), wound (*n* = 14; 12.8%), urinary tract (*n* = 6; 5.5%) and central venous catheter (CVC) (*n* = 2; 1.8%). The prevalence of CAIs and HAIs was comparable (44.1% vs. 55.9%, respectively), although this information was not available for 16 of the strains. The most frequent clinical diagnosis was CF (*n* = 32; 29.3%; *p* < 0.01 vs. other groups), followed by non-CF bronchitis or pneumonia (*n* = 15; 13.7%). Other clinical diagnoses recorded in our study population were neoplasia, wound infection, sepsis, CVC infection and intracranial injury/bleeding. Pre-exposure to antibiotics was the most prominent risk factor (*n* = 55; 50.4%; *p* < 0.0001 vs. other groups), followed by prolonged hospital/ICU stay (*n* = 17; 15.6%), chemotherapy (*n* = 12; 11.0%) and assisted ventilation (*n* = 11; 10.1%). Antibiotic therapy led to a favorable outcome in 68.3% of patients.

In reviewing the 76 putative pathogenic strains (i.e., those classified as “definite”, “probable”, or “possible” pathogen), the most common isolation site was airways followed by blood (59.2% vs. 30.3%, respectively; *p* < 0.001), whereas in the strains with a “definite” pathogen role, the most frequent isolation site was blood (60.8%; *p* < 0.01 vs. other groups). The infection was acquired with a comparable prevalence in community and hospital settings (65.8% vs. 63.5%, respectively). The previous administration of antibiotics was a risk factor, being significantly more prevalent among putative pathogenic strains compared to the non-pathogenic ones (70.9% vs. 29.1%, respectively; *p* < 0.0001). The administration of antibiotics cured the infection in most of cases (65.6% vs. 34.4%, respectively for cleared and uncleared outcomes; *p* < 0.001) and the same trend was observed for infections caused by strains classified as “definite” (69.6% vs. 26.0%, respectively; *p* < 0.001) and “probable” (44.4% vs. 26.7%, respectively; *p* < 0.05).

### 3.2. The Biofilm Forming Ability Is Highly Preserved in S. maltophilia, Although Strains with a Definite Etiological Role, Particularly Those From Blood, Show a Higher Efficiency

Biofilm forming ability in clinical *S. maltophilia* strains, as indicated by the results of MTP assay performed, is highly preserved (Figure 1). Namely, a great majority of the strains tested were recognized as biofilm producers (100 out of 109, 91.7%). However, the high variability observed in biofilm biomass values (OD_492_ range: 0.150–3.089; coefficient of variation: 77.7%) suggests the existence of streaking strain-to-strain differences in the biofilm formation efficiency (Figure 1).

The biofilm formation capability was comparable between non-pathogen strains and those with a putative pathogenic role (93.9% vs. 90.8%, respectively), as indicated by the lack of significant differences in the median value of biofilm biomass (Figure 2A). The finding was the same for three groups with ascribed pathogenic relevance (Figure 2C) as well as for the strains originating from different isolation sites (Figure 2E). When the efficiency of biofilm formation was estimated through the categorization of the strains into four classes, ranging from non-producer to strong producer, a similar distribution of the classes was noted among both putative pathogen and non-pathogen strains (Figure 2B). Most strains were classified as “strong-producers” (*p* < 0.0001 vs. other classes), followed by “moderate-”, “weak-”, and “non-producers” (Figure 2B). However, it is worth noting that all strains with a “definite” etiologic role were able to form biofilm and most of them were classified as “strong-producers”, at a proportion significantly higher than that found for strains with “probable” and “possible” pathogenic role (73.9% vs. 42.4% vs. 25%, respectively; *p* < 0.05) (Figure 2D). Further, the “strong-producer” phenotype was most prevalent among strains isolated from blood (78.3%; *p* < 0.0001 vs. other groups), followed by non-CF airway (50.0%), wound (40.0%), and CF airways (32.2%) (Figure 2F).

Biofilm levels were also correlated to the setting where infection was acquired (hospital vs. community), previous administration of antibiotic therapy, administration of targeted antibiotic therapy and clinical outcome, and no significant differences in the median biofilm mass were observed (Figure 3A,C,E,G). Categorization of the strains into four biofilm classes revealed the predominance of the “strong producers, regardless of the precipitating factor analysed (Figure 3B,D,F,H). An important finding is that the strains able to form a higher biofilm amount were significantly more prevalent among HAI than CAI strains (60.6 vs. 33.3%, respectively for HAI and CAI; *p* < 0.05) (Figure 3B).

With regard to other factors possibly contributing to strain-to strain differences in biofilm formation, the “strong-producer” phenotype was significantly associated with mechanical ventilation (*p* = 0.032). Ten out of 12 (83.4%) strains recovered from respiratory specimens were classified as “strong-producers”, and one (8.3%) each for “moderate” and “non-producer” phenotypes.

### 3.3. MLST Analysis Reveals High Genetic Diversity

Twenty-eight *S. maltophilia* isolates, representatives of different pathogenic roles, isolation country and biofilm formation classes, were selected for MLST analysis (Table 2), and a comprehensive minimum spanning phylogenetic tree of the observed lineages is shown in Figure 4.

A considerable heterogeneity was observed among the *S. maltophilia* isolates analysed, as indicated by the 20 different lineages (ST4, ST24, ST26, ST28, ST77, ST93, ST172, ST174, ST183, ST186, ST211, ST219, ST233, ST239, ST249, ST265, ST295, ST319, ST321, and ST371) found (Table 2, Figure 4). The most common STs comprised four strains as follows: ST26 [BG10 (definite), BG14 (probable), 7LP (probable) and 21LP (possible) strains]; ST28 [24LP (definite), 00410 (definite), 00630 (definite) and 29LP (non-pathogen) strains]. ST4 was formed by STMA_CZ_44 (definite) and STMA_CZ_70 (non-pathogen) strains, and ST231 included 676342 (definite) and STMA_CZ_4 (possible) strains.

A decreasing trend in biofilm biomass median values was observed among multi-strain lineages (ST4 > ST26 > ST28 > ST321) but remained below the level of significance (data not shown).

Only three lineages were isolated from multiple countries: ST26 (Serbia and Spain), ST28 (Spain and Germany) and ST321 (Czech Republic and Spain) (Figure 4). It is interesting to note that STs isolated from the same country showed a low degree of similarity in the nucleotide sequence of MLST genes (Appendix A).

### 3.4. Crystal Violet Assay Is Highly Predictive in the Quantitative Analysis of Biofilm Formation

To evaluate the predictive value of the MTP colorimetric assay used in this study to measure biofilm biomass formed by *S. maltophilia*, representative strains of strong- and weak-producer biofilm classes were comparatively evaluated by SEM and CLSM analyses. The biofilm formed by the “strong-producer” STMA_CZ_44 and BG10 strains was qualitatively and quantitatively more complex than that formed by the STMA_CZ_41 strain categorized as a “weak-producer” (Figure 5 and Figure 6). The biofilm formed by STMA_CZ_44 and BG10 strains affects almost all the contact surfaces and comprised numerous cell clusters embedded into an amorphous EPS matrix (Figure 5 and Figure 6) whose composition was predominantly polysaccharidic, as shown by carbohydrate-binding Concanavalin-A stain at CLSM analysis (Figure 6). In contrast, the biofilm formed by the “weak producer” STMA_CZ_41 showed less area coverage, cellularity, and EPS amount.

### 3.5. The Antibiotic Resistance Level Is Higher Among Pathogenic Strains

The in vitro activity of seven antibiotics was evaluated by disk diffusion, and the results are shown in Appendix A. In the *S. maltophilia* strain population as a whole, the activity of minocycline and chloramphenicol (susceptibility rate of 96.3% and 90.7%, respectively) was significantly higher compared to the other antibiotics tested (at least *p* < 0.05). Ceftazidime had the lowest activity and was only effective against 47.3% of the strains. The same pattern of susceptibility rates was observed within groups of “non-pathogen” and pathogenic strains (Appendix A). However, when the strains with a definite pathogenic role were considered separately, their susceptibility rates to all antibiotics tested were similarly high, ranging from 78.3% to 100%, except for ceftazidime (susceptibility rate: 69.6%; *p* < 0.05). It is worth noting that the levofloxacin susceptibility rate of pathogenic strains (95.7%) was significantly higher compared to that seen with non-pathogenic ones (72.7%; *p* < 0.05).

Next, the level of antibiotic-resistance was assessed by referring to both the number of resistances shown by each strain, and the multi-resistance phenotypes defined according to Magiorakos et al. [26]. The prevalence of the strains that showed no resistances was significantly lower among non-pathogenic strains than those with a definite pathogenic role (21.2% vs. 47.8%, respectively; *p* < 0.05) (data not shown). Non-MDR strains (range: 84.8–95.7%) were significantly more prevalent (*p* < 0.0001) compared to both MDR and XDR phenotypes, regardless of the group considered (data not shown).

### 3.6. The Efficiency of Biofilm Formation Is Affected by Antibiotic Resistance in Pathogenic Strains Only

We evaluated the existence of a relationship between biofilm formation efficiency and antibiotic resistance, and the results are summarized in Figure 7 and Figure 8.

The strains with a pathogenic role and susceptible to colistin, ceftazidime, levofloxacin, and ticarcillin-clavulanic acid produced significantly more biofilm than the resistant counterparts did (median OD_492_; colistin: 0.437 vs. 0.614, *p* < 0.01; ceftazidime: 0.489 vs. 0.630, *p* < 0.05; levofloxacin: 0.181 vs. 0.586, *p* < 0.001; ticarcillin-clavulanic acid: 0.298 vs. 0.618, *p* < 0.01; respectively for resistant and susceptible strains) (Figure 7). In contrast, no significant differences were found in the efficiency of biofilm formation among non-pathogenic strains, regardless of the antibiotic considered (Appendix A).

The efficiency of biofilm formation was further correlated with the level of antibiotic resistance, measured as the number of antibiotic-resistances showed by each strain (Figure 8). We established that the increasing number of resistances in pathogenic strains is inversely related to the median amount of biofilm they produce (median OD_492_: 0.660 vs. 0.471 and 0.275, respectively for strains showing no resistance, two resistances and three resistances; at least *p* < 0.05) (Figure 8A). A linear regression analysis confirmed this trend since a significant negative relationship (*r*^2^ = 0.107; *p* < 0.001) between the amount of biofilm formed and the antibiotic resistance level was found (Figure 8C). In contrast, no significant correlation between biofilm production and level of antibiotic resistance was found among non-pathogen strains (Figure 8B).

The frequency of biofilm classes was not dependent on the MDR phenotype (data not shown). However, a specific trend among pathogenic strains was observed where non-MDR phenotype was significantly associated with a higher average biofilm amount compared to the MDR strains (median OD_492_: 0.574 vs. 0.216, respectively; *p* < 0.01) (Figure 8D).

### 3.7. The Preformed S. maltophilia Biofilm Is Highly Resistant to Both Cotrimoxazole and Levofloxacin

The susceptibility of mature biofilms formed by a selected set of strains representative of different biofilm formation efficiency, pathogenetic roles and sources was evaluated in vitro by measuring the MBIC and MBEC values. Cotrimoxazole and levofloxacin were selected because considered as “first-line” antibiotics for the treatment of *S. maltophilia* infection; the results are graphed in Appendix A and summarized in Table 3 and Table 4.

Overall, the planktonic-to-biofilm lifestyle transition significantly increased *S. maltophilia* resistance to cotrimoxazole and levofloxacin, regarding biofilm formation and mature biofilm. In particular, cotrimoxazole MBIC/MIC values increased significantly for all the strains tested (MBIC/MIC range: 32 to 256), while MBEC/MBC value increase ranging from >2 to >64 was noted in 93.7% of the strains tested (Appendix A, Table 3).

The inhibitory effect of levofloxacin against biofilm formation was observed in 11 out of 16 (68.7%) strains and the MBIC/MIC ratios ranged from 4 to >1024, whereas MBEC/MBC ratios ranged from 8 to >128 suggesting that, in all the strains tested, there was an increased resistance to biofilm formation (Appendix A, Table 4).

Mature biofilms formed by blood strains were significantly more resistant to levofloxacin compared to those formed by strains recovered from other samples (MBEC/MBC: ≥128 vs. 4–128, respectively) (Table 4).

## 4. Discussion

The formation of biofilms by bacteria on inert surfaces has been extensively studied, and there appears to be a direct relationship between the capability of the organisms to form a biofilm and their pathogenicity. For instance, *E. coli* isolates causing prostatitis [27] or pyelonephritis [28,29] show a higher efficiency in biofilm formation than isolates from patients with cystitis or those without a pathogenic role. Similarly, *P. aeruginosa* isolates from the infected lungs of patients with chronic obstructive pulmonary disease tend to produce more biofilm than those from blood cultures [30].

Previous data on biofilm formation by clinical *S. maltophilia* isolates are limited. In addition, most of the studies have reported retrospectively on the frequency and characterization of biofilm-producing *S. maltophilia* from collected strains [10,13,14,15,31,32] without giving any information about the relationship with clinical infection outcomes.

To the best of our knowledge, this is the first prospective, multicenter study investigating the association between the biofilm formation capability of *S. maltophilia* strains and the clinical outcomes of infections they cause, by using a representative collection of isolates stratified according to their clinical relevance. The overall results of our study strongly support a role of biofilm formation as contributing factor in the pathogenesis of *S. maltophilia* infections.

In agreement with previous studies [15,16,17,18,33], our findings showed that the ability to form biofilm is highly preserved in clinical *S. maltophilia* isolates, albeit with significant interstrain variability. Biofilm formation was noted among both the pathogenic and non-pathogenic strains, which suggests its essential role for bacterial persistence regardless of the colonization sites. However, the ability to form a higher amount of biofilm was significantly more prevalent among strains causing hospital- rather than community-acquired infections. In addition, strains with a “definite” etiological role were recognized as the most efficient at forming biofilms; namely, the proportion of “strong-producers” was significantly higher in this category as compared to the proportions recorded among strains associated with a “probable” or “possible” pathogenic role.

Further, microscopic observations indicated that clinically relevant strains form structurally complex mature biofilms within 24 h, suggesting that their transmission may be facilitated by the fast development of biofilms on clinical equipment [16].

Overall, these findings strongly suggest that the ability to produce biofilm can be considered a determinant of virulence for *S. maltophilia*, and support the role played by biofilm formation in establishing an infection.

When the efficiency of biofilm formation was correlated with the type of sample, an obvious trend emerged. The isolates from blood showed the highest capacity for biofilm formation, while those isolated from airways of CF patients produced the lowest amounts of biofilm. Our findings are consistent with an increased adherence to human bladder HTB-9 cells by blood-derived *S. maltophilia* strains compared to those from environmental or urinary sources [17].

The evidence that biofilm production by clinical strains is affected by the site of isolation might imply that regulation of the genes encoding biofilm components is modulated by environmental influences. In support of this, we found previously a significant association between a CF-associated genotype and strong biofilm formation [5], while Willsey et al. [34] observed recently that several biofilm-associated genes were induced differentially in synthetic CF sputum medium and it corresponded to an increase in aggregation and biofilm formation.

The MLST analysis performed on a representative subset revealed that all the STs were represented by one or two isolates, except for ST26 and ST28. These findings confirm those from previous studies [2,17,33,35], where *S. maltophilia* strains also showed a high heterogeneity.

Some of the STs found in this study, such as ST4, ST24, and ST77 are epidemic in that they are widely spread over the world, isolated both from human and animal sample types, as reported in PubMLST [36]. *S. maltophilia* ST28, which we found both in Spain and Germany, also circulates in Korea and the USA isolated from human samples [36]. Conversely, several STs are endemic for each country, as in the case of ST211 and ST26 (USA), ST174 (Tunisia), ST186 (Italy), ST295 (China), ST172 (Brazil), and ST371 (Australia) all isolated from human samples [36]. Other lineages we found in the present study (ST93, ST249, ST265 and ST319) are novel since they were not reported to date in PubMLST [36].

Although clonal lineages identified in our study were not equally effective in forming biofilm, it is worth noting that clonally related isolates from different laboratories did not share the same effectiveness to form biofilms. This suggests, once again, that biofilm formation is not a sequence type-specific feature, and that its expression varies substantially under different conditions. Further studies are, however, needed to confirm this hypothesis.

In search of factors possibly correlated with biofilm formation by clinical *S. maltophilia* strains, the efficiency in forming biofilm was also stratified both on demographic and clinical data. We revealed a significant association between strong biofilm formation and mechanical ventilation. This is concordant with previous studies indicating that *S. maltophilia* causes ventilator-acquired pneumonia with a mortality rate of up to 15% [37], probably related to the bacterium’s ability to grow as a biofilm on the surfaces of endotracheal tubes and tracheostomy cannula [38,39]. Further studies are warranted to evaluate whether an augmented biofilm production is expected to correlate also with the presence of other types of indwelling medical devices, such as urinary catheters and CVC.

There was no correlation between the degree of biofilm formation and other risk factors evaluated as well as with clinical outcome. This could be due to the relevant values dispersion within each group and/or to the existence of other underlying factors.

The ability of bacterial cells to transfer genes horizontally is generally enhanced within biofilm communities, thereby facilitating the spread of antibiotic resistance [40]. However, in the present study we found an inverse relationship between antibiotic resistance and biofilm formation efficiency, specifically in the case of the pathogenic strains and for some antibiotics. This has already been reported for other bacteria [41,42]. Cepas et al. [41] observed a stronger biofilm formation in *P. aeruginosa* isolates susceptible to ciprofloxacin than in their resistant counterparts, while acquisition of quinolone resistance led to a decrease in biofilm production in both *Salmonella typhimurium* and uropathogenic *E. coli* [27,43]. In another recent study it has also been observed that strong biofilm-forming *Acinetobacter baumannii* isolates exhibited low-level resistance to gentamicin, minocycline, and ceftazidime [42]. The noteworthy result obtained in our study is that inverse relationship between antibiotic resistance and biofilm formation capacity was specific for pathogenic *S. maltophilia* strains only.

The pathogenic strains susceptible to colistin, ceftazidime, levofloxacin, and ticarcillin-clavulanic acid produced significantly more biofilm than the resistant counterparts did, whereas no significant differences were found among non-pathogenic strains. This was further confirmed by the finding that the biofilm amount formed by *S. maltophilia* significantly decreases as the number of resistances to antibiotics increased, specifically among the pathogenic strains. Indeed, pathogenic *S. maltophilia* MDR isolates formed a lower average biofilm amount compared to non-MDR isolates.

It might therefore be speculated that the biofilm-forming *S. maltophilia* strains rely less frequently on intrinsic antimicrobial resistance for survival. A possible explanation is that isolates able to form abundant biofilms are not as dependent as weakly- or non-producers on the biologically costly expression of planktonic antimicrobial resistance to survive in an environment, such as hospital settings [44,45]. Consequently, *S. maltophilia* strains with an improved ability to form a biofilm could not have been selected under antibiotic pressure.

The use of trimethoprim-sulfamethoxazole (cotrimoxazole) or levofloxacin resulted in high cure rates against monomicrobial *S. maltophilia* infections, although a trend toward resistance selection with levofloxacin was observed [19]. We compared the in vitro susceptibility of planktonic and biofilm cells of selected isolates to these two antibiotics by measuring MIC and MBC values, and MBIC and MBEC values, respectively. The inhibitory effect on biofilm formation was higher for levofloxacin than cotrimoxazole (MBIC/MIC values up to 256 and >1024, respectively). This finding confirms the efficacy of fluoroquinolones in affecting *S. maltophilia* adhesion to polystyrene and, consequently, their clinical relevance for the prevention of biofilm-related infections [12].

In a way similar to other biofilm-producing bacteria, mature *S. maltophilia* biofilms were markedly more resistant to cotrimoxazole and levofloxacin compared with their planktonic counterparts. The level of resistance to levofloxacin displayed by mature biofilms formed by the isolates recovered from blood was the highest recorded. The bactericidal activity against a preformed biofilm was compromised in a comparable way, as indicated by the MBEC/MBC values up to 128 for both antibiotics. However, it is worth noting that activity of levofloxacin against mature biofilms formed by bloodborne isolates resulted significantly affected compared to cotrimoxazole.

Overall, these data indicate that biofilm-grown *S. maltophilia* cells express an increased resistance to antimicrobial agents compared with planktonic cells, regardless of the microbiologic and clinical features of the strain (i.e., biofilm formation efficiency, pathogenetic role and source). The increased resistance of bacteria embedded in the biofilm might result in treatment failure and a recurrent infection in patients infected with biofilm-forming isolates. Clinicians should therefore manage infections caused by biofilm-forming isolates very carefully. However, no standard therapeutic guidelines for the management of *S. maltophilia* biofilm-associated infections are currently available.

The main strength of our study relies on its prospective design, along with the inclusion of a large number of patients from multiple geographically distanced medical centers, which avoid introducing bias or minimizing generalizability. Additionally, the included patients met specific criteria for infection, namely those proposed by the CDC [22], and were one-blind reviewed to limit overstating clinical outcomes.

However, some limitations should be noted. First, we used a simplified laboratory assay for quantifying biofilm formation in vitro, namely the well-described and frequently published MTP method. Although microscopic analysis demonstrated that MTP assay is highly indicative of the biofilm amount formed on polystyrene, there is increasing evidence that in vitro assays do not necessarily accurately represent the in vivo biofilm-forming ability of the isolate [46,47]. Assessing biofilm formation on microtiter plates under selected growth conditions may indeed not reflect the circumstances and gene expression during an infection in a living host. Yet, there is no standard laboratory protocol to assess in vivo biofilm formation.

Second, *S. maltophilia* can be involved in polymicrobial infections, mainly with *P. aeruginosa* and *Staphylococcus aureus* [44] which may influence the clinical outcome. During polymicrobial infections, *P. aeruginosa* elicited significantly higher *S. maltophilia* load in murine bronchoalveolar lavages and lung tissue [48]. In addition, when grown in mixed biofilm with *S. maltophilia*, over-expression of alginate by *P. aeruginosa* might be responsible for the protection of *S. maltophilia* against antibiotics [3]. According to our data, co-infecting pathogens were present in a minority of cases (<10%). In addition, we considered the patient cured only when all infecting bacteria were eradicated, and only these patients were included in the final analysis.

## 5. Conclusions

The highly conserved ability to form biofilm showed by *S. maltophilia* clinical strains, along with the significant increase in resistance to antibiotics during the planktonic-to-biofilm transition, clearly indicate that the capability for biofilm development may enable *S. maltophilia* to maintain its ecological niches as commensal microorganisms and that it can be a major virulence factor with important clinical repercussions.

Indeed, this study demonstrated that biofilm formation could be critical in the pathogenesis of *S. maltophilia* deep infections and revealed several practical implications worthy of future study. First, the strong association between biofilm formation and the site of infection makes testing for a biofilm an important tool in discriminating true bacterial infection from contamination. Second, as high-level biofilm formation seems to be fundamental to invasive *S. maltophilia* infections, tailoring diagnostic procedures to improve detection and quantification of biofilms should be considered. Third, since the biofilm-embedded cells exhibited a much higher antibiotic resistance than planktonic counterparts, the antibiotic susceptibilities of biofilm-forming *S. maltophilia* isolates should be interpreted cautiously. Furthermore, the testing of antibiotic susceptibility would conceivably be more accurately performed in a biofilm state.

The lack of correlation of *S. maltophilia* infection outcome with in vitro biofilm formation might be due to the limitations of the study-design and/or indicate that biofilm formation is one factor that, in combination with others, contributes to the development of human infections. Further studies with larger cohorts of patients and isolates are required to improve our understanding of clinical impact of *S. maltophilia*’s ability to form biofilms.

## Figures and Tables

**Figure 1 microorganisms-09-00049-f001:**
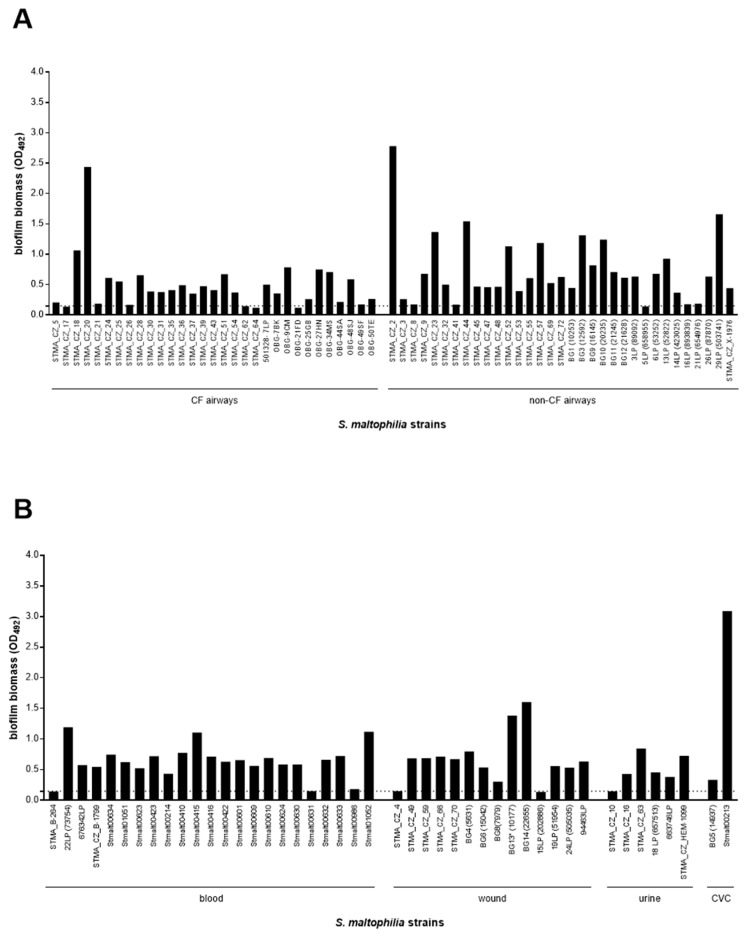
In vitro biofilm formation by 109 *S. maltophilia* strains grouped according to a clinical specimen type. The strains were collected from different sites at diagnostic laboratories of several countries (STMA: Czech Republic; BG: Serbia; Stmalt: Germany; LP: Spain; OBG: Italy): (**A**) airways, from patients with or without cystic fibrosis (CF); (**B**) blood, wound, urine, and central venous catheter (CVC). Biofilm biomass formation was evaluated using microtiter plate assay. Results are expressed as the mean optical density read at 492 nm (OD_492_), with the horizontal dotted line indicating the OD cut-off value for biofilm formation (OD_492_: 0.150).

**Figure 2 microorganisms-09-00049-f002:**
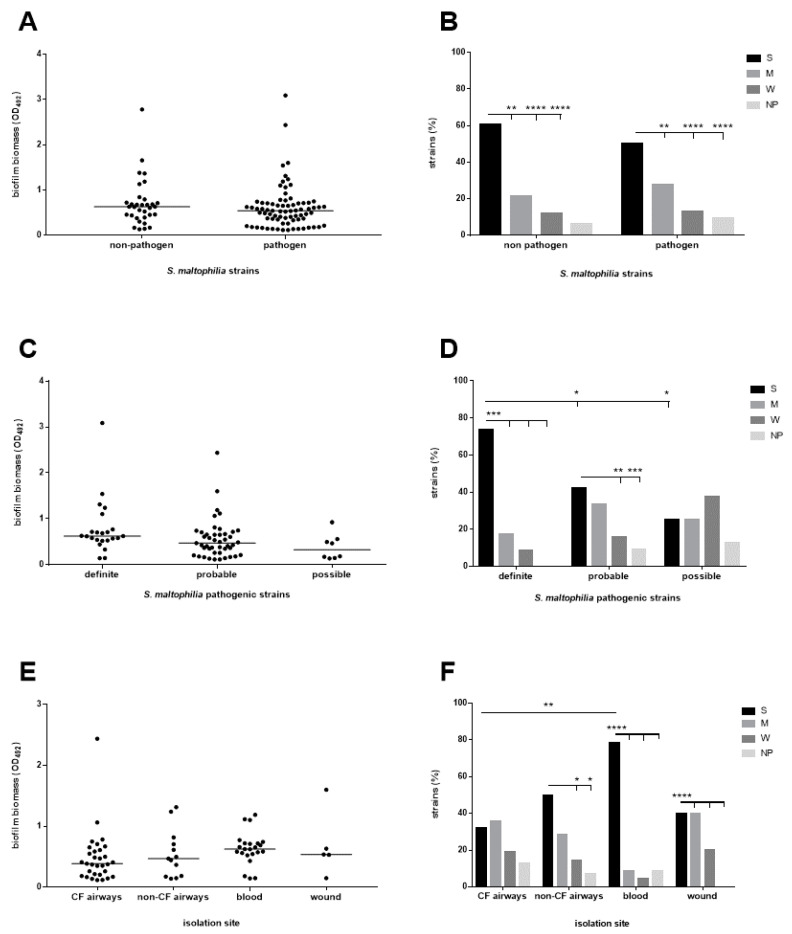
Biofilm formation by *S. maltophilia* strains and microbiological traits. Biofilm biomass formation was evaluated using MTP assay. Each strain was classified as: (i) strong—(S), moderate—(M), weak—(W) or non-producer (NP) based on the biofilm biomass value and according to Stepanović et al. [24]; (ii) pathogen (definite, probable, possible) or non-pathogen according to the CDC guidelines [22]. (**A**,**C**,**E**) Each dot shows the mean biofilm biomass (OD492 value; *n* ≥ 6), with the horizontal line showing the median value of each distribution. No statistically significant differences were found using a Kruskal-Wallis test followed by a Dunn’s multiple comparison post-test. (**B**,**D**,**F**) the percentage of strains belonging to each group. * *p* < 0.05, ** *p* < 0.01, *** *p* < 0.001, and **** *p* < 0.0001, Fisher’s exact test.

**Figure 3 microorganisms-09-00049-f003:**
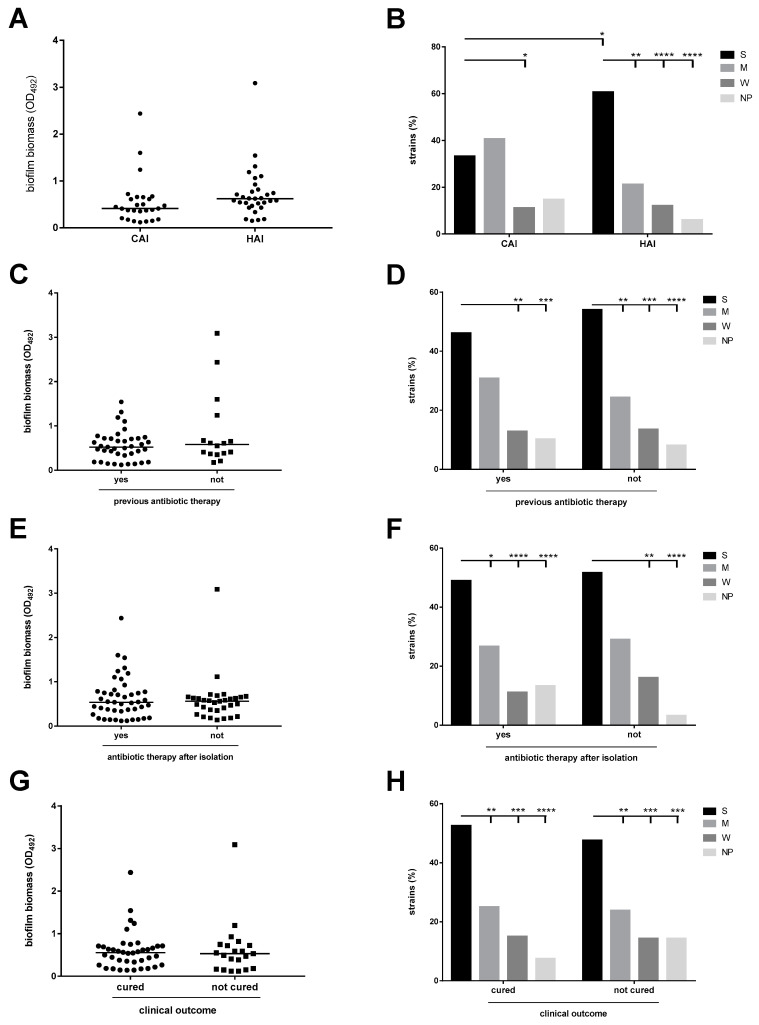
Biofilm formation by *S. maltophilia* strains: stratification on infection source, antibiotic therapy, and clinical outcome. Biofilm biomass formation was evaluated using a MTP assay. Biofilm values were stratified on: (**A**,**B**) infection source (community-acquired or hospital-acquired infection); (**C**,**D**) administration of antibiotic therapy before *S. maltophilia* isolation; (**E**,**F**) administration of antibiotic therapy after *S. maltophilia* isolation; and (**G**,**H**) clinical outcome after antibiotic therapy. (**A**,**C**,**E**,**G**) Each dot shows the mean OD_492_ value, with the horizontal line showing the median value of each distribution. (**B**,**D**,**F**,**H**) Each strain was classified as: strong—(S), moderate—(M), weak—(W) or non-producer (NP) based on the biofilm biomass value and according to Stepanović et al. [24]. * *p* < 0.05, ** *p* < 0.01, *** *p* < 0.001, and **** *p* < 0.0001, Fisher’s exact test.

**Figure 4 microorganisms-09-00049-f004:**
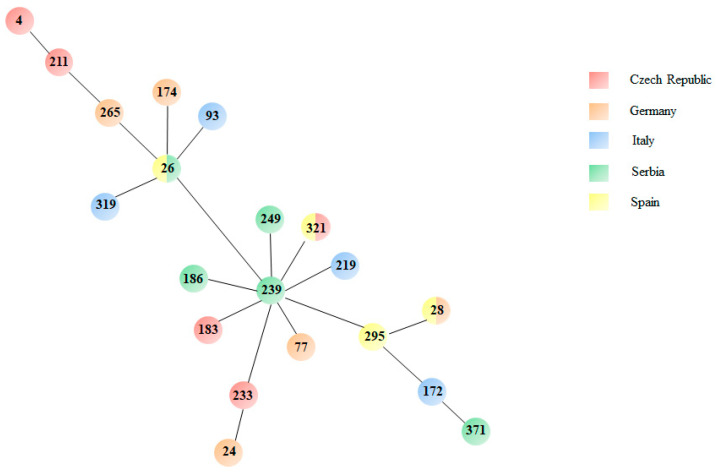
Comprehensive phylogenetic tree of *S maltophilia* strains selected for MLST analysis. Minimum spanning tree obtained using PHLOViZ Online software. The sequence types are shown in the circles, while isolation countries are indicated with different colors.

**Figure 5 microorganisms-09-00049-f005:**
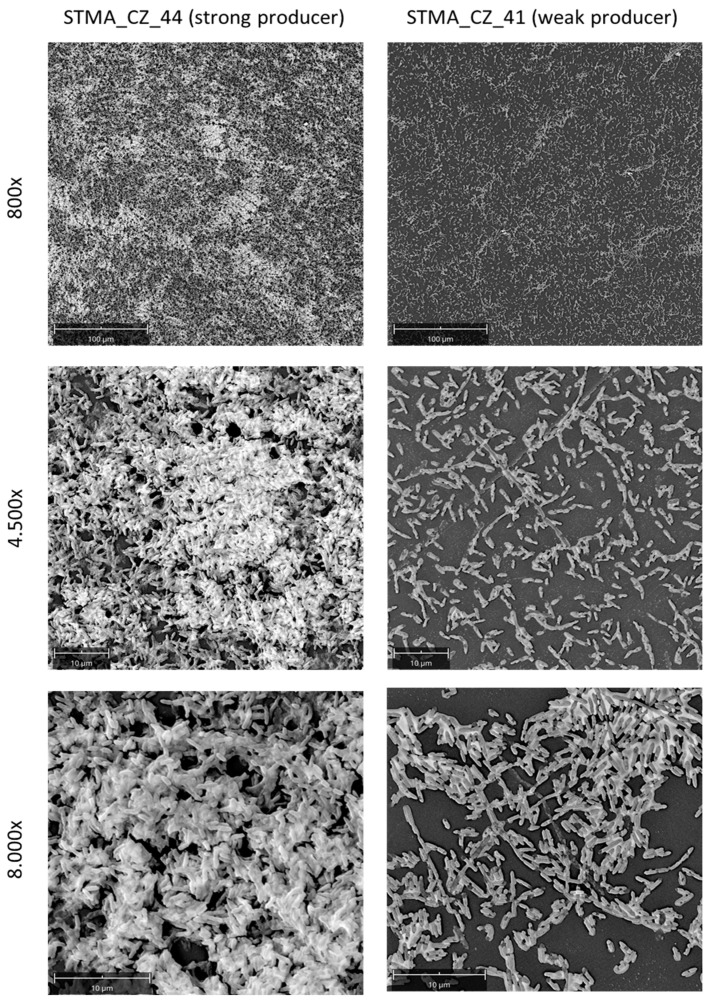
Ultrastructure analysis of *S. maltophilia* biofilm: scanning electron microscopy. The biofilm could form onto polystyrene coupons for 24 h at 37 °C by *S. maltophilia* STMA_CZ_44 and STMA_CZ_41 strains, representative of “strong” and “weak” biofilm-producer classes, respectively. Magnifications: 800×, 4500×, 8000×.

**Figure 6 microorganisms-09-00049-f006:**
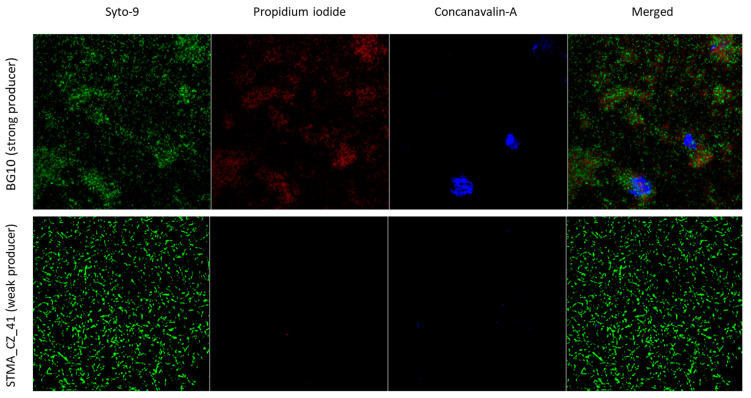
Ultrastructure analysis of *S. maltophilia* biofilm: confocal laser scanning microscopy. The biofilm was allowed to form in a µ-Dish for 24 h at 37 °C by *S. maltophilia* BG10 and STMA_CZ_41 strains, respectively representative of “strong” and “weak” biofilm producer classes. Biofilm sample was then stained using BacLight Live/Dead kit: Syto-9 tags live cells (green fluorescence); propidium iodide tags dead cells (red fluorescence); and Concanavalin-A tags extracellular polymeric substance (blue fluorescence). Merged: co-localization of Syto-9, propidium iodide and Concanavalin-A. Magnification: 1000×.

**Figure 7 microorganisms-09-00049-f007:**
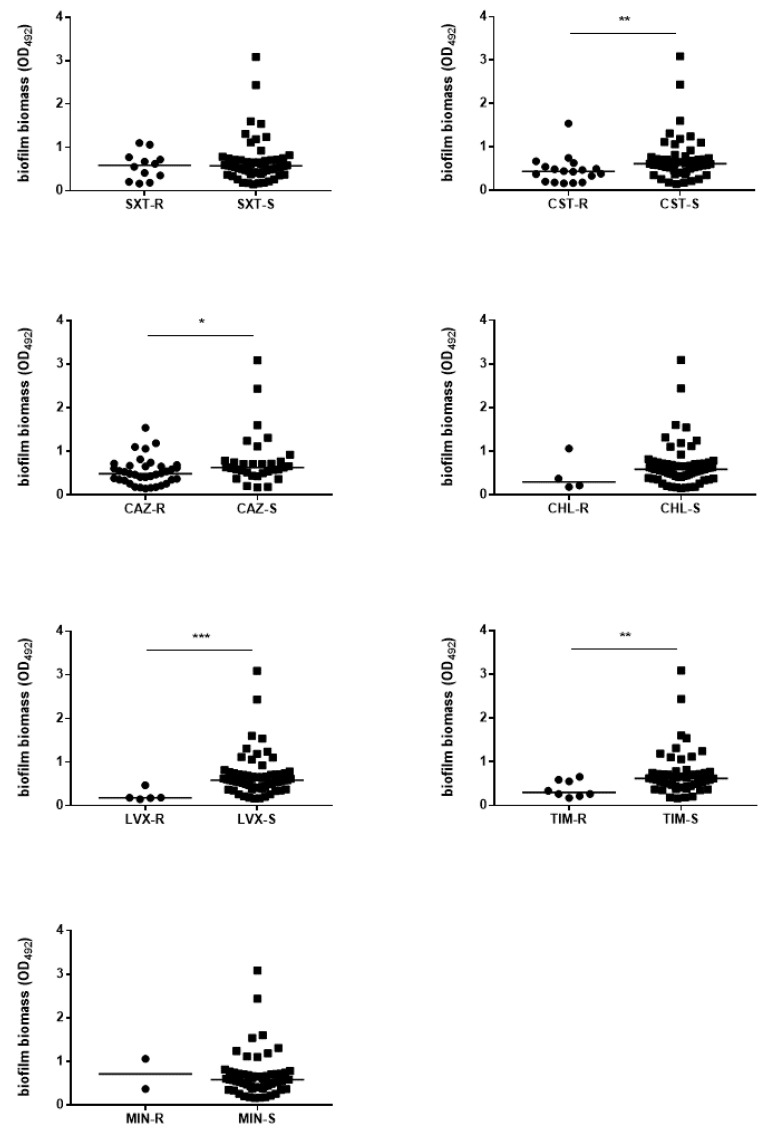
Biofilm formation by *S. maltophilia* pathogenic strains: stratification on the susceptibility phenotype. Pathogenic strains—defined according to the CDC guidelines [22]—were tested for biofilm formation in a 96-well polystyrene microtiter plate by crystal violet assay after a 24 h-incubation at 37 °C. Results were stratified on the resistance (R) or susceptibility (S) to each antibiotic tested (SXT, trimethoprim-sulfamethoxazole; CST, colistin; CAZ, ceftazidime; CHL, chloramphenicol; LVX, levofloxacin; TIM, ticarcillin-clavulanic acid; MIN, minocycline) evaluated by the disk diffusion technique. Each dot shows the mean OD_492_ value, with the horizontal line indicating the median value of each distribution. * *p* < 0.05, ** *p* < 0.01, *** *p* < 0.001, Kruskal-Wallis test followed by a Dunn’s multiple comparison post-test.

**Figure 8 microorganisms-09-00049-f008:**
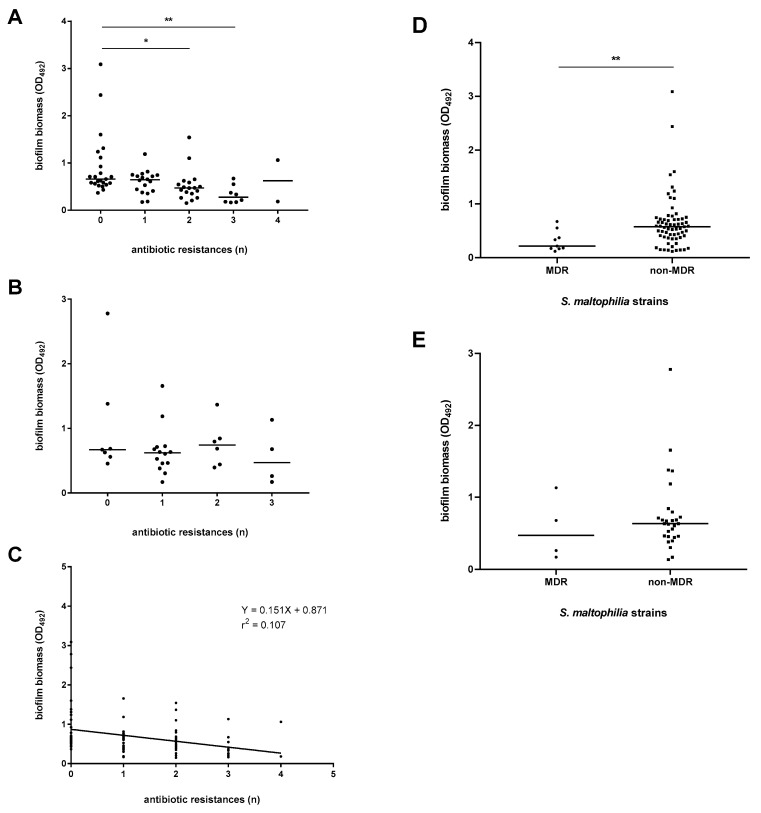
Biofilm formation by *S. maltophilia* strains: stratification on the number of antibiotic resistances. Biofilm biomass formed after a 24 h-incubation at 37 °C was measured by microtiter plate assay. Results were stratified on the strains’ pathogenic role, assigned according to the CDC guidelines [22]: (**A**,**D**) pathogen (definite + probable + possible); (**B**,**E**) non-pathogen. (**C**) Linear regression between biofilm formation and the number of resistances (*r*^2^ = 0.107; *p* < 0.001). Each dot shows the mean OD_492_ value, with the horizontal line showing the median value of each distribution. * *p* < 0.05, ** *p* < 0.01, Kruskal-Wallis test followed by a Dunn’s multiple comparison post-test (**A**,**B**), or Mann-Whitney test (**D**,**E**).

**Table 1 microorganisms-09-00049-t001:** Microbiological and clinical characteristics of *S. maltophilia* strains and patients enrolled in the present study.

No. (%) of *S. maltophilia* with the Following Etiological Role: ^a^					
	Overall (*n* = 109)	Definite (*n* = 23)	Probable (*n* = 45)	Possible (*n* = 8)	Non-Pathogen (*n* = 33)
**Isolation Site**					
Airways (CF ^b^ and non-CF)	64 (58.7)	(21.7)	34 (75.5)	6 (75.0)	19 (57.6)
Blood	23 (21.1)	14 (60.8)	8 (17.7)	1 (12.5)	
Wound	14 (12.8)	2 (8.7)	2 (4.4)	1 (12.5)	9 (27.3)
Urine	6 (5.5)		1 (2.2)		5 (15.1)
CVC ^c^	2 (1.8)	2 (8.7)			
**Source of Infection ^d^**					
Community-acquired	41 (44.1)	3 (7.3)	22 (53.6)	2 (4.9)	14 (34.1)
Hospital-acquired	52 (55.9)	18 (34.6)	11 (21.1)	4 (7.7)	19 (36.5)
**Patient**					
Age (mean ± SD)	43.8 ± 28.3	58.8 ± 25.5	27.5 ± 22.4	65.7 ± 14.1	49.8 ± 28.8
Gender (Male)	63 (57.8)	15 (65.2)	16 (35.5)	6 (75.0)	24 (72.7)
**Clinical Diagnosis**					
CF	32 (29.3)		32 (71.1)		
Non-CF airways infection	15 (13.7)	5 (21.7)	3 (6.6)	5 (62.5)	2 (6.0)
Neoplasia ^e^	8 (7.3)	3 (13.0)	2 (4.4)	1 (12.5)	2 (6.0)
Wound infection	6 (5.5)	1 (4.3)	1 (2.2)		4 (12.1)
Sepsis	6 (5.5)	2 (8.7)	3 (6.6)		1 (3.0)
CVC infection	4 (3.6)	3 (13.0)	1 (2.2)		
Intracranial injury/bleeding	4 (3.6)	1 (4.3)			3 (9.1)
Other ^f^	34 (31.2)	8 (34.8)	4 (4.4)	2 (25.0)	20 (60.6)
**Risk Factors**					
Previous antibiotic therapy	55 (50.4)	15 (65.2)	18 (40.0)	6 (75.0)	16 (48.5)
Prolonged hospital/ICU stay	17 (15.6)	3 (13.0)	3 (6.6)	1 (12.5)	10 (30.3)
Chemotherapy	12 (11.0)	4 (17.4)	3 (6.6)	1 (12.5)	4 (12.1)
Mechanical ventilation	11 (10.1)	3 (13.0)	1 (2.2)		7 (21.2)
**Outcome after Antibiotic Therapy ^g^**					
Cleared	40 (65.6) ^g^	16 (69.6)	20 (44.4)	4 (50.0)	NA ^h^
Uncleared	21 (34.4) ^g^	6 (26.0)	12 (26.7)	3 (37.5)	NA

^a^ The etiological role of each strain was defined according to the CDC guidelines [22]; ^b^ CF, cystic fibrosis. ^c^ CVC, central venous catheter. ^d^ Information available for only 93 out of 109 strains. ^e^ Including acute myeloid lymphoma, bronchial/pulmonary neoplasms, and oropharyngeal, hepatic, or pancreatic carcinomas. ^f^ Including craniosynostosis, cardiac insufficiency, cardiac surgery, cholestasis, epilepsy, cervical vertebral fracture, onychomycosis, diabetes, osteomyelitis, endocarditis, etc. Clinical diagnosis was not available for five patients. ^g^ Information available only for 61 out of 76 patients. Percentage values were calculated on 61 putative pathogenic strains only (19 “definite”, 34 “probable”, 8 “possible”). ^h^ NA, not applicable.

**Table 2 microorganisms-09-00049-t002:** *S.**maltophilia* strains chosen for Multilocus Sequence Typing (MLST) analysis. Strains were selected as representative for different etiological roles (according to CDC guidelines [22]), biofilm formation classes (according to Stepanović et al. [24]), and isolation countries.

Strain	Sequence Type	Etiological Role	Biofilm Class	Isolation Country
STMA_CZ_44	4	Definite	Strong	Czech Republic
STMA_CZ_B1799	233	Definite	Strong	Czech Republic
STMA_CZ_25	183	Probable	Strong	Czech Republic
STMA_CZ_30	211	Probable	Moderate	Czech Republic
STMA_CZ_4	321	Possible	Weak	Czech Republic
STMA_CZ_70	4	Non-pathogen	Strong	Czech Republic
BG1	239	Definite	Moderate	Serbia
BG10	26	Definite	Strong	Serbia
BG11	249	Definite	Strong	Serbia
BG9	371	Probable	Strong	Serbia
BG14	26	Probable	Strong	Serbia
BG8	186	Non-pathogen	Weak	Serbia
94463	295	Definite	Strong	Spain
676342	321	Definite	Strong	Spain
24LP	28	Definite	Moderate	Spain
7LP	26	Probable	Moderate	Spain
21LP	26	Possible	Weak	Spain
29LP	28	Non-pathogen	Strong	Spain
00623	77	Definite	Moderate	Germany
00410	28	Definite	Strong	Germany
00416	265	Definite	Strong	Germany
00610	174	Definite	Strong	Germany
00630	28	Definite	Strong	Germany
01052	24	Probable	Strong	Germany
9CM	319	Probable	Strong	Italy
7BK	93	Probable	Moderate	Italy
48SJ	219	Probable	Strong	Italy
50TE	172	Probable	Moderate	Italy

**Table 3 microorganisms-09-00049-t003:** In vitro susceptibility of preformed biofilms by *S. maltophilia* to cotrimoxazole. Minimum Biofilm Inhibitory Concentration (MBIC) and Minimum Biofilm Eradication Concentration (MBEC) values, expressed as µg/mL, were measured for a set of strains representative of different pathogenetic roles and sources. Statistically significant MBEC/MBC values (significance set at >2) are in bold. According to Stepanović et al. [24], all strains are “strong-producers”, except for BG12 (“moderate-producer”).

Strain	Source	MIC	MBIC	MBIC/MIC	MBC	MBEC	MBEC/MBC
Definite Pathogen							
STMA_CZ44	non-CF airways	1/19	32/608	**32**	16/304	>32/608	**>2**
BG3	non-CF airways	0.25/4.75	>32/608	**>128**	1/19	>32/608	**>32**
BG10	non-CF airways	0.25/4.75	>32/608	**>128**	0.5/9.5	>32/608	**>64**
BG11	non-CF airways	0.25/4.75	>32/608	**>128**	4/76	>32/60	**>8**
00422	blood	1/19	32/608	**128**	4/76	>32/608	**>8**
00610	blood	2/38	>32/608	**>16**	8/152	>32/608	**>4**
00624	blood	2/38	>32/608	**>16**	8/152	>32/608	**>4**
00630	blood	2/38	>32/608	**>16**	8/152	>32/608	**>4**
01052	blood	1/19	>32/608	**>32**	8/152	>32/608	**>4**
Probable Pathogen							
BG9	non-CF airways	1/19	32/608	**32**	8/152	>32/608	**>4**
BG12	non-CF airways	0.25/4.75	64/1216	**256**	4/76	64/1216	**16**
BG14	wound	0.125/2.375	32/608	**256**	0.5/9.5	>32/608	**>64**
STMA_CZ20	CF airways	1/19	32/608	**32**	4/76	>32/608	**>8**
Possible pathogen							
00213	blood	0.5/9.5	>32/608	**>64**	4/76	>32/608	**>8**
00634	blood	1/19	>32/608	**>32**	8/152	>32/608	**>4**
Non-pathogen							
STMA_CZ2	non-CF airways	1/19	>32/608	**>32**	32/608	>32/608	>1

**Table 4 microorganisms-09-00049-t004:** In vitro susceptibility of preformed biofilms by *S. maltophilia* to levofloxacin. Minimum Biofilm Inhibitory Concentration (MBIC) and Minimum Biofilm Eradication Concentration (MBEC) values, expressed as µg/mL, were measured for a set of strains representative of different pathogenic roles and sources. Statistically significant MBEC/MBC values (significance set at >2) are in bold. According to Stepanović et al. [24], all strains were “strong-producers” except for BG12 (“moderate-producer”).

Strain	Source	MIC	MBIC	MBIC/MIC	MBC	MBEC	MBEC/MBC
Definite Pathogen							
STMA_CZ44	non-CF airways	1	1	1	1	32	**32**
BG3	non-CF airways	0.25	>256	**>1024**	0.5	4	**8**
BG10	non-CF airways	0.5	4	**8**	1	>128	**>128**
BG11	non-CF airways	0.5	2	**4**	2	128	**64**
00422	blood	1	8	**8**	2	>256	**>128**
00610	blood	1	4	**4**	4	512	**128**
00624	blood	0.5	2	**4**	1	>128	**>128**
00630	blood	1	8	**8**	1	128	**128**
01052	blood	1	256	**256**	2	>256	**>128**
Probable Pathogen							
BG9	non-CF airways	2	4	2	2	256	**128**
BG12	non-CF airways	1	1	1	1	128	**128**
BG14	wound	1	16	**16**	1	>128	**>128**
STMA_CZ20	CF airways	1	1	1	1	8	**8**
Possible pathogen							
00213	blood	2	>256	**>128**	2	>256	**>128**
00634	blood	1	2	2	2	>256	**>128**
Non-Pathogen							
STMA_CZ2	non-CF airways	1	4	**4**	1	>128	**>128**

## Data Availability

The data presented in this study are available on request from the corresponding author. The data are not publicly available due to privacy restrictions.

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
