# Peer review of "Biofilm Formation among Stenotrophomonas maltophilia Isolates Has Clinical Relevance: The ANSELM Prospective Multicenter Study"

_microorganisms, 2020, doi:10.3390/microorganisms9010049_

Round 1

Reviewer 1 Report

Comments to the manuscript ID microorganisms-1010590

The research article submitted by Pompilio et al. describes the relationship between the ability of Stenotrophomonas maltophilia isolates to develop biofilm and their clinical outcome in human patients. The topic of this study seems interesting and relevant to the area of infectious diseases and public health in general. The authors used a collection of over 100 clinical isolates and performed biofilm assay, MLST and various antimicrobial susceptible assays. They found out that the biofilm formation was positively correlated with mechanical ventilation, whereas a negative correlation was observed between antimicrobial resistance and biofilm development.

My feeling about this manuscript is that the authors used too many classifications and comparisons of only 100 and something isolates. Many of these comparisons do not yield any important observation. There are a lot of overlaps.

The authors on several occasions mentioned that this collection represents a geographically diverse group of clinical isolates, but still do not exploit this feature in their study. It would be beneficial for the reader to see the genetic relatedness of these selected STs and even more important to show an association of these clinical isolates with other international isolates and clonal complexes. From the presented data, it seems that isolates from Spain formed two distinct STs with isolates from Serbia ST26 and Germany ST28. Are these STs emerging, pandemic and what is their history?

Minor comments:

The authors did not include line numbers, so it is hard to point them out!

Reviewer 2 Report

The researcher investigated one hundred nine isolates collected from various geographical origins. They found that most strains were able to form biofilm, and they were mostly bloodborne pathogens. Finally, they concluded that the biofilm formation by S. maltophilia contributes to the pathogenesis of infections.

I found the manuscript interesting, it may also be interesting for the readers.

Reviewer 3 Report

In this paper, authors described the association between biofilm formation and clinical outcomes of Stenotrophomonas maltophilia infections collected from five European hospitals. The authors demonstrated the ability of such strain to resistance to antibiotics during the planktonic-to-biofilm transition highlighting the consequent clinical repercussions and this represents a great a novelty in the point of the continuous growing awareness of the efficacy and development of novel antimicrobials.

However, there are some issues that should be addressed before publication:

- firstly, authors guidelines should be carefully checked, including line numbers…

- “It has the propensity to cause a plethora of opportunistic infections in humans, whereas it is mostly associated with respiratory infections [1].”  Please revise the meaning of the sentence.

- “However, since these studies were only aimed to investigate phenotypic and genotypic characteristics or virulence traits, the evidence that the formation of biofilms by S. maltophilia contributes to the clinical course of diseases is still lacking”. Changes in language and style should be performed.

- “…European hospitals and were evaluated for their ability to form a biofilm as well as their antibiotic resistance and genetic heterogeneity.” Changes in language and style should be performed.

- “…microbiological and clinical data aimed to determine their relationships.” Changes in language and style should be performed.

- “…patients met specific criteria for infection”. Which criteria? Could you please specify?

- “…and were independently reviewed to limit overstating clinical outcomes.” How were independently reviewed? Blindly? One or more reviewers? Were these well-trained?

- I believe authors exceed the allowed number of self-citations in the manuscript, they need to reduce the number of these.

Reviewer 4 Report

The manuscript by Pompilio et al, entitled "Biofilm Formation among Stenotrophomonas maltophilia Isolates Has Clinical Relevance: the ANSELM Prospective Multicenter Study"  This is an excellent study to establish a relationship between biofilm formation ability, clinical significance and virulence level. 

Comments:

- Can the authors construct a tree of the S. maltophilia isolates based on the MLST to show the clade of sequence type?

Round 2

Reviewer 1 Report

The manuscript has been improved. 

Author Response

Also on behalf of all co-authors, I would like to thank the Reviewer for appreciating our efforts to improve the manuscript.

Best regards,

Giovanni Di Bonaventura